# Endophytic Bacterial Community, Core Taxa, and Functional Variations Within the Fruiting Bodies of *Laccaria*

**DOI:** 10.3390/microorganisms12112296

**Published:** 2024-11-12

**Authors:** Kaixuan Zhang, Xin Chen, Xiaofei Shi, Zhenyan Yang, Lian Yang, Dong Liu, Fuqiang Yu

**Affiliations:** 1The Germplasm Bank of Wild Species & Yunnan Key Laboratory for Fungal Diversity and Green Development, Kunming Institute of Botany, Chinese Academy of Sciences, Kunming 650201, China; zhangkaixuan@mail.kib.ac.cn (K.Z.); shixiaofei@mail.kib.ac.cn (X.S.); yangzhenyan@mail.kib.ac.cn (Z.Y.); yanglian2@mail.kib.ac.cn (L.Y.); 2College of Life Sciences, University of Chinese Academy of Sciences, Beijing 101408, China; 3College of Life Sciences, Northwest Agriculture and Forestry University, Yangling 712100, China; chenxin723@nwafu.edu.cn

**Keywords:** *Laccaria*, mycorrhizal fungi, endophytic bacteria, community composition, ecological function

## Abstract

Macrofungi do not exist in isolation but establish symbiotic relationships with microorganisms, particularly bacteria, within their fruiting bodies. Herein, we examined the fruiting bodies’ bacteriome of seven species of the genus *Laccaria* collected from four locations in Yunnan, China. By analyzing bacterial diversity, community structure, and function through 16S rRNA sequencing, we observed the following: (1) In total, 4,840,291 high-quality bacterial sequences obtained from the fruiting bodies were grouped into 16,577 amplicon sequence variants (ASVs), and all samples comprised 23 shared bacterial ASVs. (2) The *Allorhizobium*-*Neorhizobium*-*Pararhizobium*-*Rhizobium* complex was found to be the most abundant and presumably coexisting bacterium. (3) A network analysis revealed that endophytic bacteria formed functional groups, which were dominated by the genera *Allorhizobium*-*Neorhizobium*-*Pararhizobium*-*Rhizobium*, *Novosphingobium*, and *Variovorax*. (4) The diversity, community structure, and dominance of ecological functions (chemoheterotrophy and nitrogen cycling) among endophytic bacteria were significantly shaped by geographic location, habitat, and fungal genotype, rather than fruiting body type. (5) A large number of the endophytic bacteria within *Laccaria* are bacteria that promote plant growth; however, some pathogenic bacteria that pose a threat to human health might also be present. This research advances our understanding of the microbial ecology of *Laccaria* and the factors shaping its endophytic bacterial communities.

## 1. Introduction

Fungi function as symbionts, often acting as microbial endosymbionts [1] and forming cooperative relationships with microbes within their mycelium [2]. They create distinct ecological niches that attract specialized members of the bacterial community [3,4,5]. Studies have shown that a diverse range of microorganisms inhabit macrofungi [6,7], with some promoting spore germination and enhancing mycelial health and growth [3].

Mushrooms, a type of edible macrofungi, are economically valuable crops known for their high protein content and low levels of carbohydrates and fat. The Food and Agriculture Organization of the United Nations (FAO) recommends mushrooms as a healthy food, and they are widely popular among consumers worldwide. Microorganisms, particularly bacteria, play a crucial role in the growth and production of edible mushrooms [8]. Bacteria from the phyla Actinobacteria, Firmicutes, Proteobacteria, and Bacteroidetes have been found to directly or indirectly promote the growth of edible mushrooms. Dominant genera within these phyla include *Pseudomonas* [9,10,11], *Streptomyces* [12], and *Bacillus* [13,14]. These bacterial genera fulfill various physiological and ecological roles, including substrate degradation; supplying essential nutrients and minerals; and releasing nitrogen (N), phosphorus (P), and potassium (K) to support mycelial growth and mycorrhizal formation [8,15,16].

Mycorrhiza are symbiotic relationships between soil fungi and higher plants [17]. Most terrestrial plants form these symbiotic relationships with mycorrhizal fungi, including ectomycorrhizal fungi (EMF) [18]. In nature, EMF must effectively colonize plant roots to form mycorrhiza [19]. The ability of fungi to colonize young plant roots and form mycorrhiza is influenced not only by abiotic factors such as soil pH, fertility, humidity, and temperature but also by biotic factors like microorganisms [1]. The term “Mycorrhization Helper Bacteria” (MHB) was introduced by Garbaye [20], who suggested that MHB colonize plant roots and indirectly promote plant growth. The primary auxotrophic bacteria associated with mycorrhiza include *Agrobacterium*, *Burkholderia*, *Pseudomonas*, *Bacillus*, *Paenibacillus*, and *Streptomyces* [21,22].

EMF are important symbionts that contribute significantly to nutrient cycling within ecosystems [23,24]. EMF recruit specific microbial populations from the soil by supplying various carbon sources, such as alginate, mannitol, and organic acids [25]. It should be noted that the *Laccaria*, being typical ectomycorrhizal fungi, recruit specific bacterial populations from the soil bacterial microbiota by providing different metabolites. *Laccaria bicolor* S238N accumulates high levels of alginate in the hyphae, which has the effect of chemoattracting and promoting the growth of MHB strain *Pseudomonas fluorescens* BBc6R8 [26]. Strain HB44 was able to grow avidly at the expense of the glycerol liberated by the *Lyophyllum* sp. strain Karsten [27]. As ectomycorrhizal mycorrhiza form, these recruited bacteria colonize the mycorrhiza, establishing a distinct endophytic bacterial community [28,29,30]. Research indicates that these recruited bacteria aid in mycorrhiza formation by stimulating spore germination, mycelial branching, and the development of ectomycorrhiza [21,22,31]. Moreover, these bacteria form a three-way symbiosis with EMF and the host plant, enhancing the cycling of nutrients like N and P within the ecosystem [32,33,34].

*Laccaria* Berk. & Broome is a monophyletic genus comprising around 120 species within the family Hydnangiaceae, found in both temperate and tropical regions worldwide [33,35,36,37]. To date, 32 *Laccaria* taxa have been reported in China [33,36,38,39,40,41,42,43]. The genus *Laccaria* is both economically and ecologically significant, with many species being edible and widely consumed in markets [44]. Additionally, the genus *Laccaria* is a significant ectomycorrhizal fungus that forms symbiotic relationships with a diverse range of conifer and angiosperm hosts [45]. They can be cultured stably in laboratories and are extensively utilized in ECM research. Previous research on *Laccaria* has mainly centered on discovering new species and understanding its role as an EMF, including its ability to form mycorrhiza with host plants and boost their resistance [45,46].

Unique compartments within EMF act as distinct niches for microbes, each containing its own microbiome and functional genes from both taxonomic and functional perspectives [3]. The microbiome mediates the processes from mycelial growth, ectomycorrhizal structure formation, and sporocarp development to spore formation in EMF [47,48]. During their development, EMF fruiting bodies come into contact with the soil and host a wide diversity of microorganisms. Thus, EMF fruiting body development is shaped by the soil microbiome and various abiotic factors such as annual climate variations and soil chemical composition [49,50]. Geographic factors can lead to variations in EMF morphology and their endophytic microbiomes [51]. However, the endophytic microbiomes of EMF, especially in edible species, have not been thoroughly studied. Variations in geographic location, fungal compartments, and genotype influence the composition of endophytic microbes. Therefore, this study aimed to (i) examine the abundance, diversity, composition, and core taxa of endophytic bacteria in *Laccaria* across different environments and (ii) assess the roles of geographic location, fungal genotype, and mushroom compartments (cap and stipe) in shaping the variation in the *Laccaria* endophytic bacteriome.

To achieve this, we collected seven wild *Laccaria* species from five locations: *L. aurantia* (three sites), *L. moshuijun* (two sites), *L.* sp. (two sites), *L. amethystina* (one site), *L. pallidorosea* (one site), *L. parva* (one site), and uncultured *Laccaria* (one site). We used quantitative real-time PCR, 16S rRNA gene sequencing, and network analysis to investigate variations in the *Laccaria* endophytic bacteriome. We hypothesized that (i) despite geographic variation, different species of *Laccaria* have similarly endophytic bacterial communities due to the their similar affinities (belonging to the same genus) and vertical spread of the bacterial microbiota and that they would share a core bacterial group (H1), and (ii) while endophytic bacteria in fruiting bodies perform various functions that support host fungus and their symbiotic plants’ growth and development, they would have similar functional predictions but vary in the abundance of functional genes based on geographic location, host fungal genotype, and mushroom compartment (cap and stipe) (H2).

## 2. Materials and Methods

### 2.1. Sampling Method and Storage

*Laccaria* fruiting bodies were collected from four locations in Yunnan Province, southwestern China, during the fruiting season of August–September 2023 (Table 1, Appendix A). For three independent biological replicates, we harvested 3–5 fruiting bodies from each *Laccaria* sample at three distinct forest sites within the same provenance. Only mature fruiting bodies were selected, excluding immature and decaying mushrooms. In the field, we used sterile scalpels to cut carefully the fruiting bodies, ensuring no soil was included. The freshly fruiting bodies were collected using sterile latex gloves to prevent contamination, placed in separate sterile zip-lock bags, and then brought back to the laboratory or transported in iceboxes within 12 h. To prevent being crushed in transit, each zip-lock bag was wrapped individually in foil. After being brought back (transported back) to the laboratory, the samples were processed on the same day as soon as possible; samples that could not be processed on the same day were first stored in a refrigerator at 4 °C and then were fully processed within 2 days.

### 2.2. Species Identification of Laccaria Samples

DNA was extracted from the newly collected specimens using a DNA extraction kit (Cat#DP305-03, Tiangen Biotech Corporation, Beijing, China, Appendix A). The internal transcribed spacer (ITS) region was amplified using the universal primer combination ITS1/ITS4 (Tsingke, Kunming, China) [52]. An amount of 30 µL polymerase chain reaction (PCR) reaction mix contained 15 µL 2 × PCR Mix, 0.75 µL of each primer (5 µM), 2 µL template DNA, and 11.5 µL ddH_2_O. The PCR conditions were initial denaturation at 95 °C for 3 min, followed by 33 cycles of denaturation at 95 °C for 30 s, annealing at 52 °C for 1 min, and elongation at 72 °C for 8 min. The PCR products were then sequenced by Tsingke Biotech Corporation (Beijing, China) using the same primers.

The total length of the amplified ITS sequences was about 680 bp, and after trimming off the sequences of poor quality at the beginning and the end, the good-quality sequences were about 600 bp, which were used for the subsequent phylogenetic analysis. The sequences were aligned with GenBank data using BLAST (2.13.0) and further processed with MEGA 11.0. The phylogenetic tree (Figure 1) was constructed using the neighbor-joining (NJ) method with RAxML v.7.2.6 [53]. *Mythicomyces corneipes* (Fr.) Redhead & A.H. Sm. was chosen as the outgroup based on the studies of Popa et al. [39] and Wilson et al. [45]. Statistical support for the Neighbor-Joining (NJ) tree was evaluated with nonparametric bootstrapping using 1000 replicates.

Following the comparison and construction of the phylogenetic tree, 11 samples were classified into 7 distinct species. All sequences have been deposited in GenBank under accession number PP898128–PP898138.

### 2.3. Laccaria Fruiting Body Separation

Field-fresh fruiting bodies were collected using sterile latex gloves to prevent contamination. The selected fruiting bodies were individually wrapped in aluminum foil, placed in separate sterile bags, and transported to the laboratory in iceboxes within 12 h. Prior to fruiting body separation, surface debris was removed using a fine bristle brush. The fruiting bodies were then surface-cleaned with sterilized milli-Q water and dried with sterilized absorbent paper. In the laboratory, the stipe base was cut with a sterile scalpel. Only the cap and stipe were analyzed to avoid contamination from soil microbes adhered to the stipe base. Subsequently, the fruiting bodies were separated under axenic conditions using sterile scalpels into two parts based on *Laccaria*’s distinct fruiting body structures: C = cap, S = stipe. Each cap and stipe were shredded with sterile scissors and divided into three separate sub-samples. The sub-samples were immediately placed in sterile self-sealing bags (60 mm × 85 mm) and stored at −20 °C until bacterial analysis.

### 2.4. Fruiting Body Bacteriome and Quantitative Real-Time PCR Analysis

*Laccaria* fruiting bodies were homogenized and pulverized using an MM400 Mixer Mill (Retsch, Haan, Germany). DNA was extracted using a Power Soil DNA kit (12888, MoBio R, Carlsbad, CA, USA). The V5–V7 region of the bacterial 16S rRNA gene was amplified with the following primers: F: AACMGGATTAGATACCCKG and R: ACGTCATCCCCACCTTCC [54]. PCR amplicons were purified using a gel extraction kit (OMEGA Bio-Tek, Doraville, GA, USA) and quantified with a NanoDrop 2000 (Thermo Fisher Scientific, Wilmington, DE, USA). Paired-end sequencing of the bacterial amplicons were then performed on the Illumina MiSeq-PE250 platform (Personalbio, Shanghai, China).

Raw bacterial reads were processed using the QIIME 2 pipeline. The reads were first quality-trimmed and assigned based on their barcodes. Chimeric sequences and singletons were removed. The filtered reads were then clustered into amplicon sequence variants (ASVs) with ≥99% similarity, using the taxonomically-annotated Greengenes database for 16S rRNA [55]. After quality filtering and the removal of singletons and chimeras, the high-quality 16S rRNA reads (372–380 bp) obtained from the fruiting bodies ranged from 43,042 to 106,745 per sample (mean = 72,136). These reads were clustered into ASVs with ≥97% similarity. In total, 4,840,291 high-quality bacterial sequences from the fruiting bodies were grouped into 16,577 ASVs.

All sequences have been deposited in the Sequence Read Archive (SRA) under accession number PRJNA1056774.

### 2.5. Statistical Analysis

A one-way analysis of variance (ANOVA) with Tukey’s HSD test was conducted to compare differences in bacterial abundance (both bacterial and gene copy numbers) and diversity (Shannon index) among *Laccaria* fruiting bodies from different geographic locations. Variations in the structure of the *Laccaria* fruiting body bacteriome were assessed using permutational multivariate analysis of variance (PERMANOVA) and visualized with non-metric multidimensional scaling (NMDS) plots based on Bray–Curtis distances [56]. PCoA was used to select the significant operational parameters that correlate with the ordination (latitude, longitude, and elevation).

Core and unique microbes were identified from a table of endophytic bacterial amplicon sequence variants (ASVs) and depicted using a petal diagram. Keystone microbes were identified through network analysis using a bacterial correlation matrix. Highly connected microbes were grouped into functional modules, with hub connectors calculated based on their topological roles. Module connectivity values exceeding 0.62 were considered significant, in line with previous recommendations [7,57].

The bacterial marker-gene (16S rRNA) sequences from *Laccaria* fruiting bodies were analyzed to predict their functional roles using the Functional Annotation of Prokaryotic Taxa (FAPROTAX) database (available at http://www.loucalab.com/archive/FAPROTAX (accessed on 20 October 2024)). This database enabled the functional profiling of microbial communities by linking taxonomic information to ecological functions, including nutrient cycling, metabolism, and symbiotic interactions.

## 3. Results

### 3.1. Analysis of Community Composition and Diversity of Abundant and Rare Bacteria in the Fruiting Body

Taxonomic analysis of the ASVs revealed that the endophytic bacterial communities in *Laccaria* were predominantly composed of Proteobacteria, accounting for an average of 98.93% of the total sequences. Other phyla such as Bdellovibrionota (0.36%), Actinobacteriota (0.22%), Firmicutes (0.18%), and Acidobacteriota (0.18%) were present in much lower proportions, while the remaining phyla represented only 0.13% of the total sequence abundance (Appendix A).

At the genus level, the most abundant taxa were *Allorhizobium*-*Neorhizobium*-*Pararhizobium*-*Rhizobium*, contributing 29.12% of the total sequences, followed by *Burkholderia*-*Caballeronia*-*Paraburkholderia* (14.80%), *Novosphingobium* (11.56%), *Pseudomonas* (9.79%), *Herbaspirillum* (4.16%), *Sphingomonas* (3.08%), Serratia (1.52%), *Duganella* (1.38%), *Acidisoma* (1.18%), and *Variovorax* (0.99%).

There were notable differences in abundance among samples. For example, *Allorhizobium*-*Neorhizobium*-*Pararhizobium*-*Rhizobium* was most prevalent in sample LGun (52.08%) and least abundant in JCau (2.59%). Conversely, *Burkholderia*-*Caballeronia*-*Paraburkholderia* dominated in JCau (55.40%) but had minimal presence in LKYpal (0.52%) (Figure 2).

### 3.2. Identification of the Main Drivers of Bacterial Diversity and Structure Within Laccaria Fruiting Bodies

To examine the impact of species and geographic location on the fruiting body bacteriome, we compared endophytic bacterial diversity using the Shannon diversity index. The results indicated that bacterial diversity varied significantly across different geographic locations. Specifically, the LG location exhibited significantly higher bacterial diversity compared to other locations (Figure 3A). Conversely, bacterial diversity did not vary significantly between different host fungal species (Figure 3B).

Beta-diversity was analyzed using non-metric multidimensional scaling (NMDS) (Figure 4A–C) and tested with PERMANOVA. Additionally, a variation partition analysis using the envfit function on a principal coordinate analysis (PCoA) was conducted to identify the primary factors influencing the bacterial community composition in the 11 Laccaria samples (Figure 4D). The analysis revealed noticeable variation in the fruiting body bacteriome. Among the factors examined, the host plant species (PERMANOVA test, sample size = 66, F = 7.14, *p* = 0.006 < 0.05) and geographic location (PERMANOVA test, sample size = 66, F = 5.54, *p* = 0.001 < 0.05) were identified as major drivers of bacteriome composition. In contrast, no significant differences in bacterial community composition were observed between different parts of the *Laccaria* fruiting body.

### 3.3. Core and Keystone Microbes in the Laccaria Fruiting Bodies

After examining variations in the bacteriomes of *Laccaria* fruiting bodies, we aimed to identify the core microbes that persist across different geographic conditions and host fungi. All *Laccaria* samples shared 23 core endophytic bacterial strains. Within the same *Laccaria* species, fruiting bodies from different geographic locations exhibited varying numbers of unique endophytic bacteria. For example, the number of unique bacterial ASVs in *L. aurantia* ranged from 311 in AZYau to 4174 in LGau. Similarly, within the same geographic locations, fruiting bodies from different species also had varying numbers of unique endophytic bacteria. For instance, unique bacterial ASVs in fruiting bodies from the LG location ranged from 981 in LGm to 4174 in LGau. Additionally, fruiting bodies from the LG location had a higher number of unique endophytic bacteria compared to those from other locations (Figure 5A).

In total, 23 ASVs (0.14%) from 9 bacterial families were identified as potential core members of the *Laccaria* fruiting bodies bacteriome (Figure 5D). Although their numbers were relatively small, these ASVs represented 20.52% of the total sequences across all fruiting bodies (Figure 5B). These ASVs were classified into the following families: *Burkholderiaceae* (8.20%), *Rhizobiaceae* (6.69%), *Oxalobacteraceae* (2.26%), *Comamonadaceae* (1.33%), *Xanthobacteraceae* (1.02%), *Sphingomonadaceae* (0.61%), *Beijerinckiaceae* (0.28%), *Acetobacteraceae* (0.11%), and unclassified *Rhizobiales* (0.01%) (Figure 5C).

Given that all fruiting bodies were from the same fungal genus and collected from a single province, we performed a bacterial network analysis to better understand the key taxa present (Figure 6). The network analysis showed that the bacteria did not exist in isolation but co-occurred in clusters. Modules 7 (blue) and 2 (red) were particularly interconnected and formed the largest module, highlighted by a light blue ellipse (Figure 6). This network was primarily dominated by the genera *Allorhizobium-Neorhizobium-Pararhizobium-Rhizobium*, *Novosphingobium*, and *Variovorax* (Figure 6B).

### 3.4. Bacteriome Function

To better understand the endophytic bacteriome, we clustered all the selected fungal species and analyzed the primary functional types present in the endophytic bacterial microbiota of their fruiting bodies (Appendix A).

Figure 7 illustrates the variations in the relative abundance of key ecological functions in the endophytic bacterial community of *Laccaria* fruiting bodies, considering factors such as fruiting body compartment, geographic location, and host fungal genotype. Most of these functions are characterized by chemoheterotrophy and aerobic chemoheterotrophy, which together account for over 70% of the total functional abundance (Appendix A). Besides these, N-cycling is also a prominent function, highlighted by high levels of ureolysis and N fixation (Figure 7). The primary ecological functions, such as chemoheterotrophy and N-cycling, are significantly affected by geographic location and host fungal genotype, rather than the fruiting body compartment of the fungi (Figure 7).

Additionally, we compared the top 10 most abundant endophytic bacterial genera in *Laccaria* fruiting bodies (Appendix A) with the plant-beneficial and -harmful bacteria database [58]. Our analysis demonstrated that among the 10 most abundant endophytic bacterial genera, eight were either beneficial or partially beneficial to plants (Figure 8). The *Allorhizobium*-*Neorhizobium*-*Pararhizobium*-*Rhizobium* complex had the highest abundance, reaching 27.73%. In comparison, only 3 of the top 10 bacterial species were harmful to plants, and their abundances were very low.

To identify endophytes within *Laccaria* fruiting bodies that could potentially impact human health, we made a reference to the food macro-genome database [59]. Their study identified 43 commonly occurring single-genome-bins (SGBs) in food and humans. Among the top 10 most abundant endophytic bacterial species in *Laccaria* samples (as shown in Appendix A), *Lactococcus lactis* (with an abundance of 3.08%) was the only one among the 43 SGBs.

## 4. Discussion

### 4.1. Laccaria Endophytic Bacterial Abundance and Composition

This study identified a diverse range of bacterial taxa in the fruiting bodies of *Laccaria*, spanning 11 phyla, 40 classes, 120 orders, 222 families, 496 genera, and 1065 species. Initially, we hypothesized (H1) that the endophytic bacteria would be similar across *Laccaria* species due to their shared fungal host. However, our findings revealed significant variations in bacterial abundance among different *Laccaria* species. Notably, L. *aurantia* exhibited a higher abundance of *Burkholderia*-*Caballeronia*-*Paraburkholderia* compared to other species, while it had the lowest abundance of *Allorhizobium*-*Neorhizobium*-*Pararhizobium*-*Rhizobium*. These results indicate that differences in host fungi influence the composition and abundance of their endophytic bacterial communities.

ECM associations significantly affect the abundance of endophytic bacteria. The establishment of ECM symbiosis alters the physiological and biochemical processes within the host plant, resulting in a modified microenvironment in the root system. This environment directly impacts the growth and proliferation of endophytic bacteria [28]. For example, ECM associations can enhance nutrient uptake, creating a more favorable environment for certain endophytic bacteria and increasing their abundance. On the other hand, changes in root exudates due to ECM interactions may selectively promote or inhibit specific bacterial populations [27,31,60]. Moreover, the immune responses and defense mechanisms influenced by ECM can affect the composition and abundance of endophytic bacteria. A robust plant immune system may restrict the growth of certain bacterial species while enabling others to thrive [61,62]. The intricate interactions between ECM and the host plant led to substantial changes that ultimately influence the abundance and dynamics of endophytic bacterial communities.

The group of microorganisms known as *Allorhizobium*-*Neorhizobium*-*Pararhizobium*-*Rhizobium* (ANPR) is commonly found in EMF [30,32,63,64]. These bacteria play several crucial roles in EMF ecosystems. They facilitate nutrient cycling by aiding in the uptake and transformation of essential elements like N, P, and K, thereby improving the nutritional status of both the EMF and its host plants. ANPR may also influence the growth and development of EMF by secreting substances that promote hyphal elongation or fruiting body formation, thus supporting the health of the *Laccaria* fruiting body [64]. Furthermore, ANPR are abundant in various parts of EMF, including fruiting bodies, rhizomorphs, and the mycosphere of *Cantharellus cibarius*, a widely distributed EMF. They help protect the EMF from pathogens by producing antimicrobial compounds or competing for resources and space [65]. ANPR can also regulate gene expression within EMF, affecting physiological processes and responses to environmental stresses such as drought or extreme temperatures. Overall, ANPR’s presence in EMF indicates a complex and dynamic interaction with significant implications for the health, functionality, and ecological roles of both the fungi and the associated plant systems.

Alongside the ANPR complex, the *Laccaria* fruiting bodies also harbored high abundances of *Burkholderia*-*Caballeronia*-*Paraburkholderia*, *Novosphingobium*, and *Pseudomonas*. The association of *Burkholderia*-*Caballeronia*-*Paraburkholderia* and *Pseudomonas* with saprotrophic mushrooms is well documented [66,67]. Members of the *Burkholderiaceae* family (*phylum Proteobacteria*), which are often sensitive to acidic conditions [68,69], may use saprotrophic fruiting bodies as refuges from low pH environments [70,71]. *Novosphingobium* is known for its ability to degrade aromatic compounds [72,73] and is a promising bioremediation agent for aromatic-contaminated environments. During the colonization of *Populus trichocarpa* by the ECM fungus *Laccaria bicolor*, there was a notable shift in aromatic acids [74], suggesting that *Novosphingobium* may play a role in the colonization process of *Laccaria* fungi.

The genus *Pseudomonas* is referred to as “fungiphiles” [75,76]. According to Warmink et al., fungiphiles are selected based on their ability to utilize organic substrates present in fungal exudates [75]. The high relative abundance of *Pseudomonas* in *Laccaria* fruiting bodies is attributed to the high N and P content typical of saprotrophic fungi fruiting bodies [77].

We found that geographic difference has a significant impact on ECM (*Laccaria* in our case) endophytic bacteria. The relative abundance of *Laccaria* endophytic bacteria varies across different habitats (LKY, JC, LG, and AZY). These geographic variations lead to distinct environmental conditions, including differences in climate, soil composition, and resource availability, which in turn influence the types and abundances of ECM endophytic bacteria [2,5,32]. In general, factors such as average humidity during the summer season and precipitation patterns can have a significant impact on the growth and distribution of bacteria. High humidity and regular precipitation can create a conducive environment for the growth of many bacteria, as they require moisture for their growth and reproduction.

Among the four sampling sites in this study, although Lijiang Alpine Botanic Garden (LABG) has a lower mean annual precipitation than Yunnan Academy of Forestry and Grassland (YAFG), when the samples were collected, it was during the rainy season at LABG, with the highest humidity and a slightly lower temperature. This may have led to more bacteria entering the interior of the fungal fruiting bodies to colonize. In addition, YAFG has the second highest mean annual precipitation, which contributed to a high relative abundance of endophytic bacteria in the *Laccaria* fruiting bodies at this site as well.

Moreover, soil quality, including nutrient levels and pH, also plays a critical role; soils with specific minerals or pH levels can favor certain bacterial species [78,79]. Another reason for the higher relative abundance of endophytic bacteria in the fruiting body samples taken from LABG than YAFG may be due to the higher nutrient content of the humic substances (HSs) compared to the soil and, in particular, the effect of the higher N content in the HS [80]. HS is defined as a chemically and biologically degradable polyelectrolytic macromolecular substance derived from plants, animal residues, and microbial cells [80,81]. Due to its specific physical and chemical functions, such as binding metals/ions/molecules and having redox properties, it is often used to modify soil structure, maintain soil fertility, and improve the soil environment [82]. Additionally, geographical isolation can limit the movement and exchange of bacterial populations, leading to the development of distinct microbial communities [2]. Overall, these geographical factors shape the characteristics and composition of ECM endophytic bacteria, highlighting the importance of considering location in microbial studies.

### 4.2. The Key Bacterial Genera Within Laccaria Fruiting Bodies

Consistent with our first hypothesis (H1), wild *Laccaria* species shared a set of keystone microbes. Despite differences among *Laccaria* species and their geographic origins, 23 core endophytic bacterial strains were common across samples, with 14 of these belonging to the order Rhizobiales. We believe that the widespread presence and dominance of Rhizobiales in *Laccaria* species are related to their functional roles.

Rhizobiales are crucial for several important functions. They are primarily known for their role in N fixation [83], where many species form symbiotic relationships with plants, particularly legumes, to convert atmospheric N into a form usable by plants [84,85]. This process enriches soil N levels and reduces the need for synthetic fertilizers. Additionally, some Rhizobiales enhance plant growth by improving nutrient uptake and increasing resistance to biotic and abiotic stresses [86,87,88,89]. The specific functions and impacts of Rhizobiales can vary by species and ecological context, and ongoing research continues to reveal their diverse roles.

At the family level, our analysis of endophytic bacteria across various *Laccaria* species and geographic locations identified Burkholderiaceae as a prominent group among the core microbes. This family is widely distributed in diverse environments and is known to associate with various plant species [90,91], as well as with fungi [92]. *Burkholderia* species within this family are known to enhance host growth and development [91] and can tolerate and degrade a range of aromatic compounds, including monoaromatic, polycyclic aromatic, and heterocyclic compounds [93]. It is possible that certain Burkholderiaceae taxa play a role in the colonization of host plants by *Laccaria* fungi. Additionally, Burkholderiaceae bacteria are highly metabolizable and can boost host plant biomass through the production of secondary metabolites, including volatile organic compounds, and may also inhibit the growth of some pathogenic fungi [94].

### 4.3. Prediction of Ecological Function Within Laccaria Fruiting Body

The functions were predominantly characterized by chemoheterotrophy and aerobic chemoheterotrophy, which together account for over 80% of the total relative abundance.

These functions are crucial for the survival, growth, and reproduction of EMF, and they play a key role in maintaining ecological balance and nutrient cycling in the surrounding environment. For instance, the dominance of chemoheterotrophy and aerobic chemoheterotrophy in the fruiting body of EMF (*Laccaria*, in our case) can be attributed to several reasons. Firstly, chemoheterotrophs obtain energy and carbon by consuming organic compounds [95,96]. The fruiting body environment offers a rich supply of organic matter from both the surrounding substrate and the fungal own metabolic processes, making chemoheterotrophy an efficient strategy for energy acquisition. Additionally, the aerobic conditions within the fruiting body create an ideal environment for aerobic chemoheterotrophs. Aerobic metabolism is more energy-efficient than anaerobic processes [97], which supports the dominance of these organisms. Variations in microenvironments, such as oxygen levels, nutrient availability, and pH, further favor the growth and activity of chemoheterotrophs and aerobic chemoheterotrophs.

Besides chemoheterotrophy, N-cycling emerged as the second most important function within the EMF fruiting bodies, as evidenced by the high abundances of ureolysis and N fixation (Figure 7). Ureolysis, the breakdown of urea into ammonia and carbon dioxide, is essential because it (i) facilitates the recycling of nitrogenous compounds, such as urea, a common N-containing molecule, which is broken down to release N in a form usable by the EMF and associated organisms for growth and metabolic processes [98], and (ii) regulates pH within the fruiting body via ammonia production during ureolysis, creating an optimal environment for fungal enzymatic activity and physiological functions. N fixation involves converting atmospheric N into a usable form, such as ammonia or nitrate [99]. Its significance includes (i) providing N, with atmospheric N being abundant but not directly usable by most organisms and N fixation supplying essential N to the EMF, particularly in N-limited environments, and (ii) supporting ecosystem balance, with N fixation helping maintain the N balance in the ecosystem, supporting the growth of other organisms dependent on N for survival and reproduction [100,101].

The dominant ecological functions of EMF fruiting bodies, such as chemoheterotrophy and N-cycling, were significantly influenced by geographic location and host fungal genotype rather than the specific fruiting body compartments of the fungi (Figure 7). Geographic location affects factors like climate, soil composition, and nutrient availability, which in turn impact the metabolic strategies and functional adaptations of the fungi. The host fungal genotype determines the genetic traits that interact with the environment, shaping the dominant functions. Conversely, the specific fruiting body compartments of the fungi have less influence, as they are more consistent internally and less affected by external variations.

Overall, the analysis of ecological functions in EMF provides a foundational understanding of these complex relationships. However, further research is needed to uncover the precise mechanisms and interactions among these factors and their effects on the functional abundance of the endophytic bacterial community within *Laccaria* fruiting bodies.

### 4.4. Implication of Laccaria Endophytic Bacteria on Plant and Human Health

Consistent to our hypothesis (H2), endophytic bacteria in fruiting bodies also support the growth and development of host fungus’ symbiotic plants. Eight out of the top ten most abundant endophytic bacterial genera are either beneficial or partially beneficial to plants, accounting for 71.38% of the total abundance (Figure 8). The *Allorhizobium*-*Neorhizobium*-*Pararhizobium*-*Rhizobium* group (known as rhizobia), is one of the most extensively studied plant microbiota. These bacteria colonize specific plant roots, obtain mineral nutrients and energy from the plant, and fix atmospheric nitrogen to provide the plant with essential nitrogen nutrients [102]. Similarly, the *Burkholderia*-*Caballeronia*-*Paraburkholderia* group, which is known as plant growth-promoting rhizobacteria (PGPR), is capable of colonizing the root systems of at least 30 plant species and demonstrating a wide variety of growth-promoting functions such as nitrogen fixation, phosphorus solubilization, as well as biotrophic and bacteriostatic effects [103,104,105,106]. The *Novosphingobium* genus possess a broad capacity for metabolizing aromatic compounds [107,108,109], and some are potential candidates as plant growth-promoting bacteria due to their ability to alleviate salt stress [110].

As mentioned previously, EMFs can recruit specific microbial populations to colonize from the soil bacterial microbiota by providing different carbon sources. Interestingly, these recruited bacteria not only are present in the mycosphere and even adhere to the hyphae or inside the hyphae but also enter the fruiting body to form their unique endofungal bacterial microbiota during ectomycorrhizal fruiting body formation [28,30]. Experimental studies by jia et al. [34,111] have shown that in the triple symbiotic system of EMF-fruiting body endophytic bacteria–plant, the interaction of EMF and bacteria can synergistically mineralize organic phosphorus as well as synergistically solubilize inorganic phosphorus and thus, through the mycorrhizal structure, enhance phosphorus uptake by the plant.

On the other hand, as a common edible mushroom, at the level of endophytic bacterial species, the only endophytic bacterium found in *Laccaria* that is common in humans was *Lactococcus lactis*, which had a low abundance (Appendix A), was classified as Generally Recognized as Safe (GRAS) by the U.S. Food and Drug Administration (FDA) and is widely used in food fermentation and feed additives [112]. However, at the genus level, some species of *Rhizobium* [113,114,115,116] and *Burkholderia* [104,117,118,119] have been definitively reported as human pathogens. Thus, there is a potential risk of contracting a disease when handling wild *Laccaria*. However, further research is required to evaluate the pathogenic potential after cooking or steaming.

## 5. Conclusions

The study of endophytic bacteria in the fruiting bodies of the ectomycorrhizal fungus *Laccaria* offers significant insights. It reveals that the *Allorhizobium*-*Neorhizobium*-*Pararhizobium*-*Rhizobium* (ANPR) complex is highly abundant. Additionally, geographic location and host fungal genotype play crucial roles in shaping the diversity, community structure, and ecological functions of these bacteria. Different *Laccaria* species create distinct ecological niches, influencing the types and abundances of endophytic bacteria. Wild *Laccaria* species share core bacterial strains, with the order Rhizobiales possibly related to N fixation and plant growth promotion. Understanding the variations in endophytic bacteria within *Laccaria* fruiting bodies is essential for grasping microbial ecology and the intricate interactions between fungi and bacteria. In addition, a large number of endophytic bacteria in *Laccaria* are bacteria that promote plant growth, and some pathogenic bacteria that pose a threat to human health may also exist. This knowledge can promote research on ecosystem functioning and has potential applications in the fields of agriculture, environment, and health sciences.

## Figures and Tables

**Figure 1 microorganisms-12-02296-f001:**
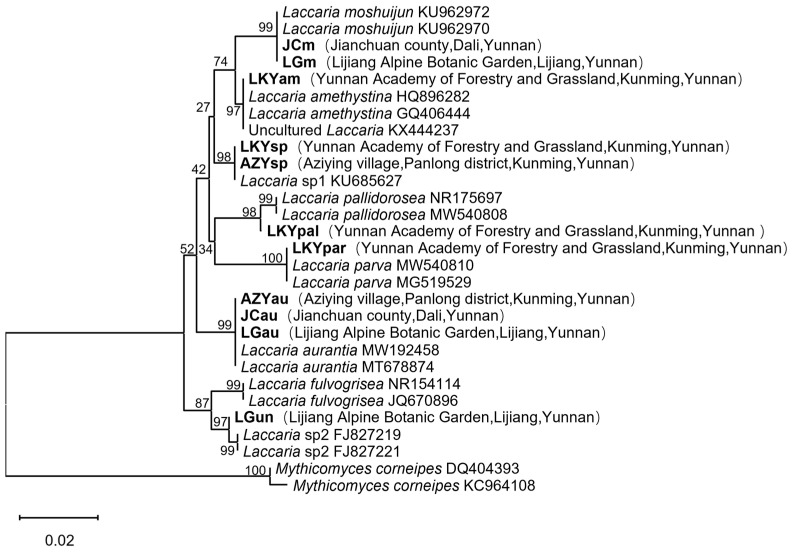
The NJ-tree of *Laccaria* was constructed using the ITS dataset. Numbers at each branch node represent bootstrap values from 1000 re-samplings. *Laccaria* sample names are shown in boldface type. Sequences of *Mythicomyces corneipes* (DQ404393 and KC964108) were included as the outgroup.

**Figure 2 microorganisms-12-02296-f002:**
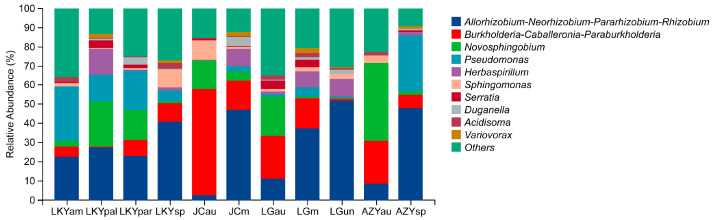
Relative abundance of top 10 endophytic bacterial genera found in the fruiting bodies of *Laccaria* samples. Relative abundance of lower and unclassified/unidentified taxa are grouped as “Others”. Values are the mean of three replicates.

**Figure 3 microorganisms-12-02296-f003:**
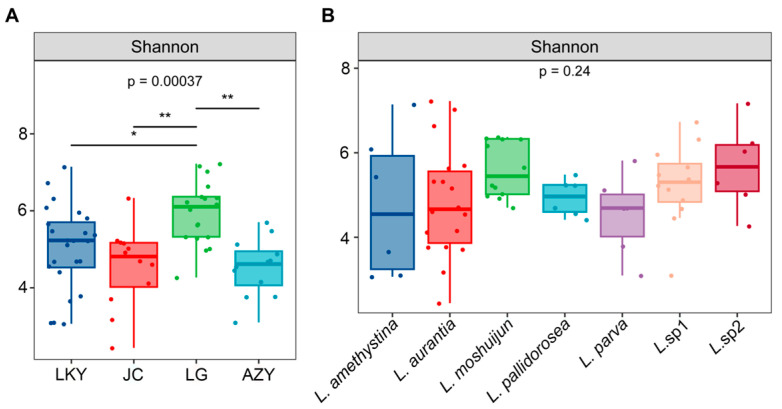
*Laccaria* endophytic bacterial diversity, as affected by geographic location (**A**) and host species (**B**). Lines over the box plot represent the *t*-test’s pairwise comparison and asterisks represent the significance of mean differences (** *p* < 0.01; * *p* < 0.05). Significantly different pairs are indicated by dark lines.

**Figure 4 microorganisms-12-02296-f004:**
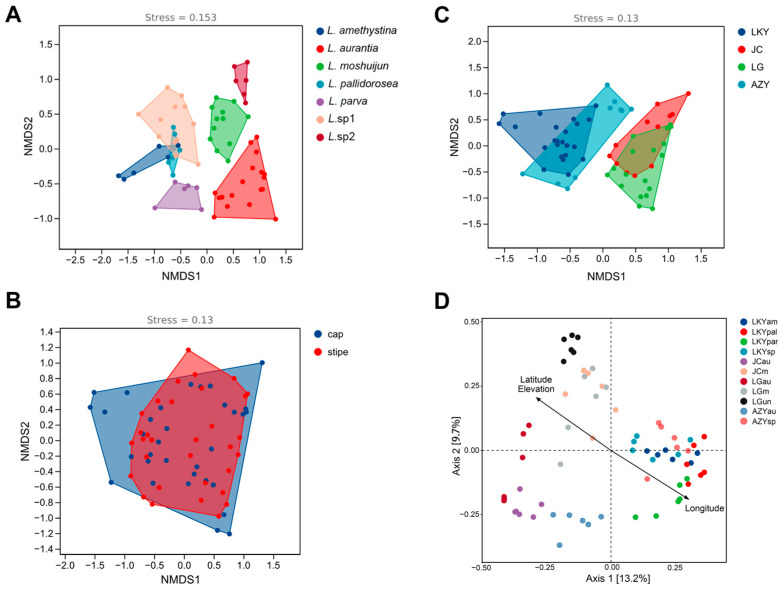
Change in *Laccaria* endophytic bacterial community structure, as affected by host fungal species (**A**), fungal fruiting body compartment (**B**), and geographic location (**C**,**D**). Endophytic bacterial community structures were indicated by non-metric multidimensional scaling plots (NMDSs) and principal coordinate analysis (PCoA) using the Bray–Curtis matrix distance.

**Figure 5 microorganisms-12-02296-f005:**
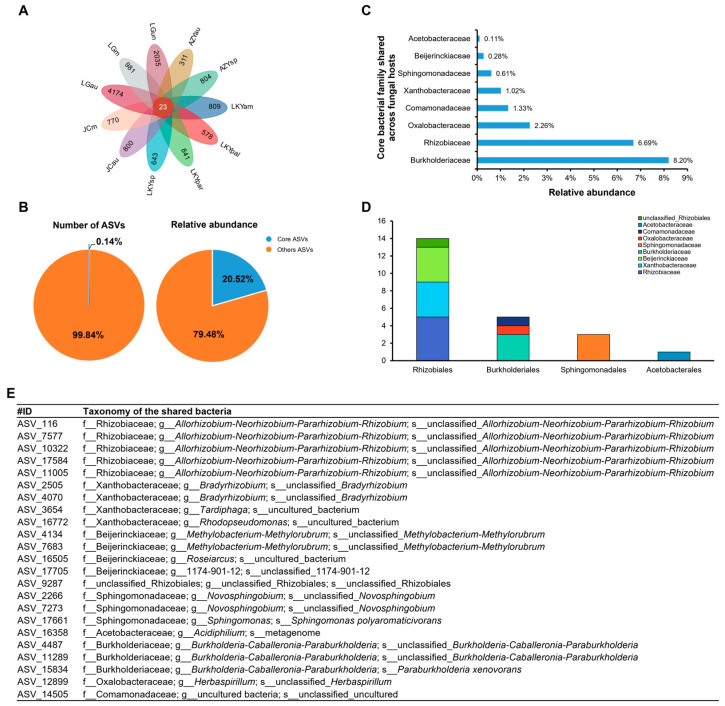
Core bacteriome of *Laccaria* samples. Petal diagram of the numbers of core and unique bacterial (**A**). Pie charts represent the relative abundance and number of ASVs detected in the core bacteriome of *Laccaria* samples (**B**). The core bacteriome of fungal hosts with their relative abundance in the whole dataset (**C**). The composition of ASVs forming the core *Laccaria* bacteriome (**D**). Core ASVs of *Laccaria* fruiting bodies (**E**).

**Figure 6 microorganisms-12-02296-f006:**
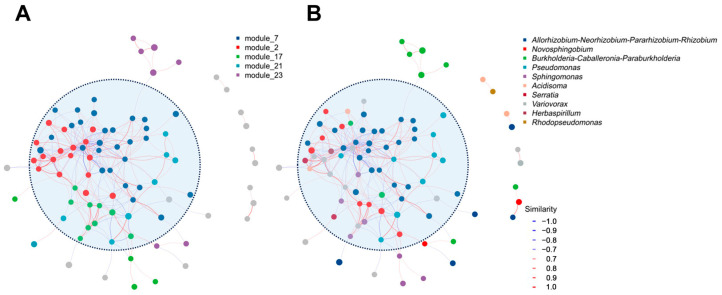
*Laccaria* fruiting body endophytic bacterial networks based on (**A**) functional modules and (**B**) dominant bacterial genera. The networks were developed using bacterial correlation matrices and ASV tables for the following *Laccaria* species: *L. amethystina* (*n* = 3), *L. pallidorosea* (*n* = 3), *L. parva* (*n* = 3), *L. moshuijun* (*n* = 6), *L.* sp1 (*n* = 6), *L. aurantia* (*n* = 9), and *L*. sp2 (*n* = 3). Nodes in the network represent the top 100 dominant bacterial taxa. The color of the lines indicates the strength of the correlation between nodes, ranging from negative (blue) to positive (red). Different colors were used to visualize modules within the network, which denote clusters of highly interconnected nodes.

**Figure 7 microorganisms-12-02296-f007:**
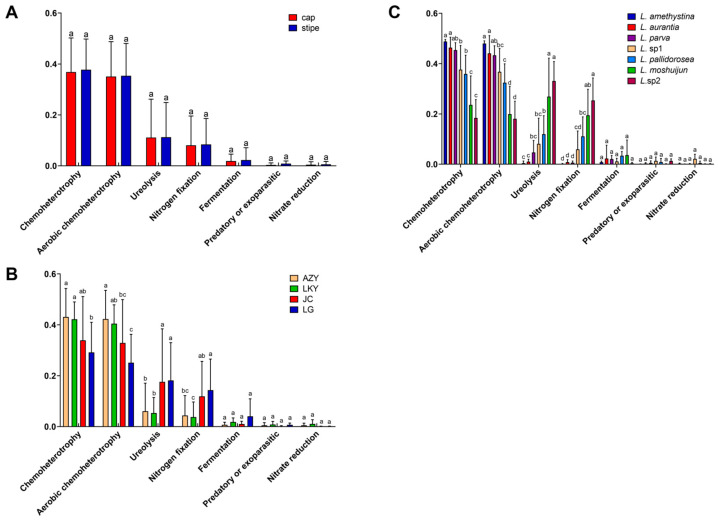
*Laccaria* endophytic bacterial community main functional abundance prediction varies with fruiting body compartment (**A**), geographic location (**B**), and the host fungal genotype (**C**) based on the FAPROTAX (1.2.1). The various upper-case letters above the bars indicate significant difference.

**Figure 8 microorganisms-12-02296-f008:**
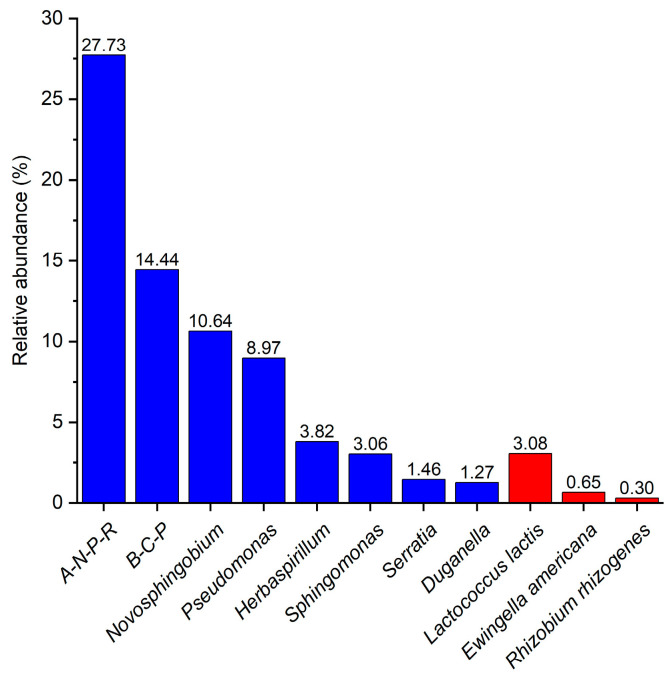
Abundant *Laccaria* endophytic functional bacteria. Plant-beneficial and -harmful taxa are shown in blue and red, respectively. Abbreviations: A-N-P-R: *Allorhizobium*-*Neorhizobium*-*Pararhizobium*-*Rhizobium*; B-C-P: *Burkholderia*-*Caballeronia*-*Paraburkholderia*. Functional taxonomy is based on published databases [58,59].

**Table 1 microorganisms-12-02296-t001:** Experimental *Laccaria* samples’ information.

Sample	Species	Location	Origins	Symbiosis Tree
LKYam	*Laccaria amethystina*	Yunnan Academy of Forestry and Grassland, Kunming	soil	*Quercus variabilis*
LKYpal	*L. pallidorosea*	Yunnan Academy of Forestry and Grassland, Kunming	soil	*Quercus variabilis*
LKYpar	*L. parva*	Yunnan Academy of Forestry and Grassland, Kunming	soil	*Quercus variabilis*
LKYsp *	*L.* sp1	Yunnan Academy of Forestry and Grassland, Kunming	soil	*Quercus variabilis*
JCau	*L. aurantia*	Jianchuan county, Dali	humic substance	*Pinus yunnanensis*
JCm	*L. moshuijun*	Jianchuan county, Dali	humic substance	*Pinus yunnanensis*
LGm	*L. moshuijun*	Lijiang Alpine Botanic Garden, Lijiang	humic substance	*Pinus yunnanensis*
LGau	*L. aurantia*	Lijiang Alpine Botanic Garden, Lijiang	humic substance	*Quercus guyavifolia*
LGun *	*L.* sp2	Lijiang Alpine Botanic Garden, Lijiang	humic substance	*Quercus guyavifolia*
AZYau	*L. aurantia*	Aziying village, Kunming	humic substance	*Quercus variabilis*
AZYsp *	*L.* sp1	Aziying village, Kunming	humic substance	*Quercus variabilis*

* Samples LKYsp, LGun, and AZYsp were not identified as species-specific, with samples LKYsp and AZYsp belonging to the same species (*L*. sp1) and sample LGun to another (*L*. sp2).

## Data Availability

The original contributions presented in the study are included in the article/Appendix A, further inquiries can be directed to the corresponding authors.

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
