# Peer review of "Endophytic Bacterial Community, Core Taxa, and Functional Variations Within the Fruiting Bodies of Laccaria"

_microorganisms, 2024, doi:10.3390/microorganisms12112296_

Round 1

Reviewer 1 Report

Comments and Suggestions for Authors

Observations: 

Lines 78-81 Enhance writing to diminish the degree of resemblance.

Lines 94-97 Consolidate the hypotheses to one or two, establishing a hypothesis encompassing all study objectives.

In the Materials and Methods section, elucidate the origins of the fungus, specifying whether it was sourced from soil or decomposing organic matter of a tree or shrub. These findings are crucial for delineating the feeding mechanisms of the fungus. The discussion section indicates that the prevalence of bacterial composition is contingent upon environmental factors and symbiotic relationships associated with nutrient availability, such as root conditions and pH; however, further data regarding the microenvironment of the collected samples are not provided. Request additional information regarding the environmental conditions of the geographical region, including the average humidity of the sampling area during the summer season. Is precipitation occurring? The year of the collection was unusual? given the variations caused by climate change, among other factors. These statistics are readily accessible and, where relevant, enhance the discussion by considering geographical conditions.

What is the maximum duration for storage at 4°C? Were they preserved in plastic receptacles or pouches?

As a reviewer, I could not obtain the methods for the DNA extraction kits. Can you incorporate them as supplementary material?

What is the quality and quantity of the DNA samples?

Author Response

Reviewer1:

Observations:

Manuscript ID: microorganisms-3255870

Lines 78-81 Enhance writing to diminish the degree of resemblance.

As suggested by the reviewer, we have revised the original sentences “Particularly, unique EMF compartments can be regarded as specific micro-niches for microbes, housing their own microbiomes and sets of functional genes from both taxonomic and functional viewpoints [3]. The development of EMF is a microbiome-mediated process involving stages from mycelial growth and ectomycorrhizal formation to fruiting body development.”

to

“Unique compartments within EMF act as distinct niches for microbes, each containing its own microbiome and functional genes from both taxonomic and functional perspectives [3]. The microbiome mediates the processes from mycelial growth, ectomycorrhizal structure formation, sporocarp development to spore formation in EMF.”

Lines 94-97 Consolidate the hypotheses to one or two, establishing a hypothesis encompassing all study objectives.

Great thanks. Based on your suggestion, we have consolidated the hypotheses to two to encompass all study objectives and adjusted accordingly later in the text:

We have revised the original old sentence “We hypothesized that: (i) the Laccaria endophytic bacteriome would show similarities due to its affinity with the host fungus (H1); (ii) wild Laccaria species would share core mi-crobes despite geographic variation, due to vertical transmission of bacterial microbiota and similarities among their hosts (H2); (iii) while endophytic bacteria in fruiting bodies perform various functions that support host fungus growth and development, they would have similar functional predictions but vary in the abundance of functional genes based on geographic location, host fungal genotype, and mushroom compartment (cap and stipe) (H3); and (iv) as an ectomycorrhizal fungus, some endophytic bacteria are beneficial to the symbiotic plants of Laccaria (H4)”.

to:

“We hypothesized that (i) despite geographic variation, different species of Laccaria have similarly endophytic bacterial communities due to the their similar affinities (belonging to the same genus) and vertical spread of the bacterial microbiota, and that they would share a core bacterial group (H1); and (ii) while endophytic bacteria in fruiting bodies perform various functions that support host fungus and their symbiotic plants’ growth and development, they would have similar functional predictions but vary in the abundance of functional genes based on geographic location, host fungal genotype, and mushroom compartment (cap and stipe) (H2)”.

In the Materials and Methods section, elucidate the origins of the fungus, specifying whether it was sourced from soil or decomposing organic matter of a tree or shrub. These findings are crucial for delineating the feeding mechanisms of the fungus. The discussion section indicates that the prevalence of bacterial composition is contingent upon environmental factors and symbiotic relationships associated with nutrient availability, such as root conditions and pH; however, further data regarding the microenvironment of the collected samples are not provided. Request additional information regarding the environmental conditions of the geographical region, including the average humidity of the sampling area during the summer season. Is precipitation occurring? The year of the collection was unusual? given the variations caused by climate change, among other factors. These statistics are readily accessible and, where relevant, enhance the discussion by considering geographical conditions.

Thanks for your professional suggestions. we have modified the contents of Table 1 by adding the origins of the samples as well as their symbiosis tree species. We also include the habitat conditions of the collection sites in the Supplementary table 1 and reorganized the order of the supplementary tables. Furthermore,we have add the discussion of differences in endophytic bacterial diversity among samples of the species from the perspective that the habitats where the samples were sampled differed, the contents are as followings:

“Among the four sampling sites in this study, the Lijiang Alpine Botanic Garden (LABG) has a lower mean annual precipitation than Yunnan Academy of Forestry and Grassland (YAFG) though. However, when the samples were collected, LABG was during the rainy season, and there had highest humidity and a slightly lower temperature, and these reasons may lead to the fact that more bacteria would enter the interior of the fungal fruiting bodies to colonize. In addition, YAFG has the second highest mean annual precipitation, which contributed to a high relative abundance of endophytic bacteria in the Laccaria fruiting bodies at this site as well.”

“Another reason for the higher relative abundance of endophytic bacteria in the fruiting body samples taken from LABG than YAFG may be due to the higher nutrient content of the humic substances (HS) compared to the soil, and in particular the effect of the higher N content in the HS [80]. HS is defined as a chemically and biologically degradable poly-electrolytic macromolecular substance derived from plants, animal residues and microbial cells. [80,81] Due to its specific physical and chemical functions, such as binding metals/ions/molecules and having redox properties, it is often used to modify soil structure, maintain soil fertility and improve the soil environment [82].”

Literature:

  • [80] Valenzuela, E.I.; Cervantes, F.J. The role of humic substances in mitigating greenhouse gases emissions: Current knowledge and research gaps. Total Environ. 2021, 750, 141677, doi:10.1016/j.scitotenv.2020.141677.
  • [81] Jin, Y.; Yuan, Y.; Liu, Z.; Gai, S.; Cheng, K.; Yang, F. Effect of humic substances on nitrogen cycling in soil-plant ecosystems: Advances, issues, and future perspectives. Environ. Manage. 2024, 351, 119738, doi:10.1016/j.jenvman.2023.119738.
  • [82] Li, Y.; Fang, F.; Wei, J.; Wu, X.; Cui, R.; Li, G.; Zheng, F.; Tan, D. Humic Acid Fertilizer Improved Soil Properties and Soil Microbial Diversity of Continuous Cropping Peanut: A Three-Year Experiment. Rep. 2019, 9, 12014, doi:10.1038/s41598-019-48620-4.

Table 1. Experimental Laccaria samples’ information.

Sample

Species

Location

Origins

Symbiosis tree

LKYam

Laccaria amethystina

Yunnan Academy of Forestry and Grassland, Kunming

soil

Quercus variabilis

LKYpal

L. pallidorosea

Yunnan Academy of Forestry and Grassland, Kunming

soil

Quercus variabilis

LKYpar

L. parva

Yunnan Academy of Forestry and Grassland, Kunming

soil

Quercus variabilis

LKYsp*

L. sp1

Yunnan Academy of Forestry and Grassland, Kunming

soil

Quercus variabilis

JCau

L. aurantia

Jianchuan county, Dali

humic substance

Pinus yunnanensis

JCm

L. moshuijun

Jianchuan county, Dali

humic substance

Pinus yunnanensis

LGm

L. moshuijun

Lijiang Alpine Botanic Garden, Lijiang

humic substance

Pinus yunnanensis

LGau

L. aurantia

Lijiang Alpine Botanic Garden, Lijiang

humic substance

Quercus guyavifolia

LGun*

L. sp2

Lijiang Alpine Botanic Garden, Lijiang

humic substance

Quercus guyavifolia

AZYau

L. aurantia

Aziying village, Kunming

humic substance

Quercus variabilis

AZYsp*

L. sp1

Aziying village, Kunming

humic substance

Quercus variabilis

Supplementary table 1. The geographic location information for the sampling sites

Sample

Location

Latitude

(N)

Longitude

(E)

Altitude

(m.a.s.l.)

Temperature (℃)

Humidity (%)

Weather

Average temperature (℃)

Average precipitation (mm)

LKYam

Yunnan Academy of Forestry and Grassland, Kunming

25°9′2″

102°44′58″

1995

26

48

cloudy

25.25-16.5

316.45

LKYpal

Yunnan Academy of Forestry and Grassland, Kunming

25°9′2″

102°44′58″

1970

LKYpar

Yunnan Academy of Forestry and Grassland, Kunming

25°9′2″

102°44′58″

1971

LKYsp

Yunnan Academy of Forestry and Grassland, Kunming

25°9′2″

102°44′58″

1966

JCau

Jianchuan county, Dali

26°31′33″

100°2′41″

3085

16

94

light rain

24.75-14.75

146.4

JCm

Jianchuan county, Dali

26°31′33″

100°2′41″

3100

LGm

Lijiang Alpine Botanic Garden, Lijiang

27°0′11″

100°12′26″

3400

17

92

moderate rain

23.25-14

196.3

LGau

Lijiang Alpine Botanic Garden, Lijiang

27°0′11″

100°12′26″

3400

LGun

Lijiang Alpine Botanic Garden, Lijiang

27°0′11″

100°12′26″

3440

AZYau

Aziying village, Kunming

25°20′8″

102°46′49″

2070

20

61

cloudy

24-15.25

133.45

AZYsp

Aziying village, Kunming

25°20′8″

102°46′49″

2060

What is the maximum duration for storage at 4°C? Were they preserved in plastic receptacles or pouches?

Thanks for the questions, we have provided a detailed sample collection, transportation, and processing sections:

Lines 113-117: “The freshly collected fruiting bodies were placed in separate plastic bags and transported to the laboratory in iceboxes within 12 hours. Each fruiting body was wrapped individu-ally in foil and stored in a refrigerator at 4℃ until processed in a laminar flow chamber. Eleven representative samples were selected for subsequent compartment separation.

Have been revised to:

“The freshly fruiting bodies were collected using sterile latex gloves to prevent contamination and placed in separate sterile zip-lock bags, then brought back to the laboratory or transported in iceboxes within 12 hours. To prevent being crushed in transit, each zip-lock bag was wrapped individually in foil. After being brought back (transported back) to the laboratory, samples are processed on the same day as far as possible. Samples that cannot be processed on the same day are first stored in a refrigerator at 4℃ and are fully processed within 2 days.”

As a reviewer, I could not obtain the methods for the DNA extraction kits. Can you incorporate them as supplementary material?

Thanks for your suggestion, we have incorporated the methods for the DNA extraction kit as supplementary material, details are as follows:

“ 1. Take about 100 mg of fresh tissue or 30 mg of dry weight tissue and add liquid nitrogen to grind it thoroughly.

  1. Transfer the milled powder quickly into a centrifuge tube pre-filled with 700μl of 65°C pre-warmed Buffer GP1, quickly invert it to mix well and place the tube in a 65°C water bath for 20 min, inverting the tube to mix the samples several times during the water bath.
  2. Add 700 μl chloroform, mix thoroughly and centrifuge at 12, 000 rpm for 5 min.
  3. Carefully transfer the upper aqueous phase from the previous step into a new centrifuge tube, add 700 μl of Buffer GP2 and mix well.
  4. Transfer the homogenized liquid to spin columns CB3 and centrifuge at 12, 000 rpm for 30 sec, discard the waste liquid.
  5. Add 500 μl of Buffer GD to the spin columns CB3, centrifuge at 12, 000 rpm for 30 sec, pour off the waste solution, and put the spin column CB3 into a collection tube (2 ml).
  6. Add 600 μl of Buffer PW to the spin column CB3, centrifuge at 12, 000 rpm for 30 sec, pour off the waste solution, and place the spin column CB3 into the collection tube.
  7. Repeat step 7.
  8. Place the spin column CB3 back into the collection tube, centrifuge at 12, 000 rpm for 2 min, pour off the waste liquid. Leave the spin column CB3 at room temperature for a few minutes to thoroughly dry the rinse solution remaining in the adsorbent material.
  9. Transfer the spin column CB3 into a clean centrifuge tube, add 50-200 μl of elution Buffer TE dropwise to the middle of the adsorbent membrane overhang, leave at room temperature for 2-5 min, centrifuge at 12, 000 rpm for 2 min, and collect the solution into a centrifuge tube.”

What is the quality and quantity of the DNA samples?

Based on your question, we have refined the description of the quantity and quality of the resulting DNA sequences, details are as follows:

“An amount of 30 µL polymerase chain reaction (PCR) reaction mix contained 15 µL 2 × PCR mix, 0.75 µL of each primer (5 µM), 2 µL template DNA, and 11.5 µL ddH2O.”

“The total length of the amplified ITS sequences were about 680 bp, and after trimming off the sequences of poor quality at the beginning and the end, the good quality sequences were about 600 bp, which were used for the subsequent phylogenetic analysis.”

Reviewer 2 Report

Comments and Suggestions for Authors

Comments to manuscript“ Endophytic bacterial community, core taxa, and functional variations within the fruiting bodies of Laccaria“ by Zhang et al.

Zhang et al. identified the endophytic bacteria and functional groups of bacteria which reside in the fruiting bodies of Laccaria species. For this study, fruiting bodies of seven species from five location were collected. After species identification, the authors perform a dissection of the endophyte communities by molecular methods, encompassing quantitative real-time PCR, and paired-end sequencing of the bacterial amplicons using the Illumina MiSeq-PE250 platform. The methods, are adequate and these experiments as well as the statistics seem to be carefully done.

The results are very interesting, disclosing a relationship between Laccaria endophytic bacterial diversity, the  geographic location of sampling and the Laccaria species, while 23 bacterial stains occur in all Laccaria species. These strains are supposed to present the core bacteriome of Laccaria and most of them have plant growth promoting prperties, according to the used database.

Nevertheless there are several concerns. The most important one is that the bacterial taxa were not correlated to the plant species, where Laccaria as an ectomycorrhizal species are symbiotic with. The bulk soil and rhizospere microorganisms were not investigated. Thus, the reasons for the variation in microbial diversity shown in Fig. 2, 3 and 4 stay unclear and the results descriptive. The authors should mentioned the plants involved in the symbiosis.

Another point are the functions of the endophytic bacteria. Why are plant growth promoting bacteria in the fruiting bodies? Please give an explanation.

Are these bacteria also associated with the mycelium? Concerning the bacteria, it is unclear, whether the mentioned functions are performed, because no tests were done. At least, such experiments and their results are not presented in the main text. Thus, this part is speculative and should either deleted or proved by experiments.

Author Response

Reviewer2:

Comments to manuscript “Endophytic bacterial community, core taxa, and functional variations within the fruiting bodies of Laccaria” by Zhang et al.

Zhang et al. identified the endophytic bacteria and functional groups of bacteria which reside in the fruiting bodies of Laccaria species. For this study, fruiting bodies of seven species from five location were collected. After species identification, the authors perform a dissection of the endophyte communities by molecular methods, encompassing quantitative real-time PCR, and paired-end sequencing of the bacterial amplicons using the Illumina MiSeq-PE250 platform. The methods, are adequate and these experiments as well as the statistics seem to be carefully done.

The results are very interesting, disclosing a relationship between Laccaria endophytic bacterial diversity, the geographic location of sampling and the Laccaria species, while 23 bacterial stains occur in all Laccaria species. These strains are supposed to present the core bacteriome of Laccaria and most of them have plant growth promoting properties, according to the used database.

We would like to express our sincere gratitude for your detailed and positive comments on our manuscript. Your recognition of the adequacy of our methods, as well as the careful execution of our experiments and statistics, is highly encouraging. We are also glad that you found our results interesting. The relationship between Laccaria endophytic bacterial diversity, sampling location, and Laccaria species is indeed an important discovery, and the identification of the 23 core bacterial strains that are potentially plant-growth- promoting is a significant outcome of this study. We believe that these findings contribute to a better understanding of the ecological roles of endophytic bacteria within Laccaria fruiting bodies. We have carefully considered your comments during the revision process and will ensure that the manuscript is further improved based on your suggestions. Once again, thank you for your valuable feedback.

Nevertheless, there are several concerns. The most important one is that the bacterial taxa were not correlated to the plant species, where Laccaria as an ectomycorrhizal species are symbiotic with. The bulk soil and rhizosphere microorganisms were not investigated. Thus, the reasons for the variation in microbial diversity shown in Fig. 2, 3 and 4 stay unclear and the results descriptive. The authors should be mentioned the plants involved in the symbiosis.

Great thanks. Based on your valuable suggestions, we have shown the samples as well as the details of the sampling locations in Table 1 and Supplementary table 1.

Table 1. Experimental Laccaria samples’ information.

Sample

Species

Location

Origins

Symbiosis tree

LKYam

Laccaria amethystina

Yunnan Academy of Forestry and Grassland, Kunming

soil

Quercus variabilis

LKYpal

L. pallidorosea

Yunnan Academy of Forestry and Grassland, Kunming

soil

Quercus variabilis

LKYpar

L. parva

Yunnan Academy of Forestry and Grassland, Kunming

soil

Quercus variabilis

LKYsp*

L. sp1

Yunnan Academy of Forestry and Grassland, Kunming

soil

Quercus variabilis

JCau

L. aurantia

Jianchuan county, Dali

organics

Pinus yunnanensis

JCm

L. moshuijun

Jianchuan county, Dali

organics

Pinus yunnanensis

LGm

L. moshuijun

Lijiang Alpine Botanic Garden, Lijiang

organics

Pinus yunnanensis

LGau

L. aurantia

Lijiang Alpine Botanic Garden, Lijiang

organics

Quercus guyavifolia

LGun*

L. sp2

Lijiang Alpine Botanic Garden, Lijiang

organics

Quercus guyavifolia

AZYau

L. aurantia

Aziying village, Kunming

organics

Quercus variabilis

AZYsp*

L. sp1

Aziying village, Kunming

organics

Quercus variabilis

Supplementary table 1. The geographic location information for the sampling sites

Sample

Location

Latitude

(N)

Longitude

(E)

Altitude

(m.a.s.l.)

Temperature (℃)

Humidity (%)

Weather

Average temperature (℃)

Average precipitation (mm)

LKYam

Yunnan Academy of Forestry and Grassland, Kunming

25°9′2″

102°44′58″

1995

26

48

cloudy

25.25-16.5

316.45

LKYpal

Yunnan Academy of Forestry and Grassland, Kunming

25°9′2″

102°44′58″

1970

LKYpar

Yunnan Academy of Forestry and Grassland, Kunming

25°9′2″

102°44′58″

1971

LKYsp

Yunnan Academy of Forestry and Grassland, Kunming

25°9′2″

102°44′58″

1966

JCau

Jianchuan county, Dali

26°31′33″

100°2′41″

3085

16

94

light rain

24.75-14.75

146.4

JCm

Jianchuan county, Dali

26°31′33″

100°2′41″

3100

LGm

Lijiang Alpine Botanic Garden, Lijiang

27°0′11″

100°12′26″

3400

17

92

moderate rain

23.25-14

196.3

LGau

Lijiang Alpine Botanic Garden, Lijiang

27°0′11″

100°12′26″

3400

LGun

Lijiang Alpine Botanic Garden, Lijiang

27°0′11″

100°12′26″

3440

AZYau

Aziying village, Kunming

25°20′8″

102°46′49″

2070

20

61

cloudy

24-15.25

133.45

AZYsp

Aziying village, Kunming

25°20′8″

102°46′49″

2060

Another point are the functions of the endophytic bacteria. Why are plant growth promoting bacteria in the fruiting bodies? Please give an explanation. Are these bacteria also associated with the mycelium? Concerning the bacteria, it is unclear, whether the mentioned functions are performed, because no tests were done. At least, such experiments and their results are not presented in the main text. Thus, this part is speculative and should either deleted or proved by experiments.

Thanks for your kind reminding. We have added explanations on why are plant growth promoting bacteria in the fruiting bodies.

we have revised the original sentence “It should be noted that the Laccaria, being ectomycorrhizal fungi, recruit bacteria from the soil which not only promotes their own growth and development but also benefits the host plant. These bacteria might provide mutual advantages to both the fungus and the host plant.

into

“As mentioned previously, EMFs can recruit specific microbial populations to colonise from the soil bacterial microbiota by providing different carbon sources. Interestingly, these recruited bacteria are not only present in the mycosphere and even adhere to the hyphae or inside the hyphae, but also enter the fruiting body to form their unique endofungal bacterial microbiota during ectomycorrhizal fruiting body formation [28,30]. Experimental studies by jia et al. [34, 110] have shown that in the triple symbiotic system of EMF-fruiting body endophytic bacteria-plant, the interaction of EMF and bacteria can synergistically mineralize organic phosphorus as well as synergistically solubilize inorganic phosphorus, thus through the mycorrhizal structure enhancing phosphorus uptake by the plant.”

Literature:

  • [28] Bai, H.Y.; Zhang, A.Y.; Mei, Y.; Xu, M.; Lu, X.L.; Dai, C.C.; Jia, Y. Effects of ectomycorrhizal fungus bolete identity on the community assemblages of endofungal bacteria. Env Microbiol Rep 2021, 13, 852-861, doi:10.1111/1758-2229.13007.
  • [30] Pent, M.; Bahram, M.; Põldmaa, K. Fruitbody chemistry underlies the structure of endofungal bacterial communities across fungal guilds and phylogenetic groups. ISME J 2020, 14, 2131-2141, doi:10.1038/s41396-020-0674-7.
  • [34] Mei, Y.; Zhang, M.; Cao, G.; Zhu, J.; Zhang, A.; Bai, H.; Dai, C.; Jia, Y. Endofungal bacteria and ectomycorrhizal fungi synergistically promote the absorption of organic phosphorus in Pinus massoniana. Plant Cell Environ. 2024, 47, 600-610, doi:10.1111/pce.14742.
  • [110] Zhang, A.Y.; Zhang, M.L.; Zhu, J.L.; Yan, M.; Xu, F.J.; Bai, H.Y.; Sun, K.; Zhang, W.; Dai, C.C.; Jia, Y. Endofungal bacterial microbiota promotes the absorption of chelated inorganic phosphorus by host pine through the ectomycorrhizal system. Microbiol Spectr 2023, 11, e00162-00123, doi:10.1128/spectrum.00162-23.

Round 2

Reviewer 2 Report

Comments and Suggestions for Authors

I have no further concerns.